# Encapsulated Phytomedicines against Cancer: Overcoming the “Valley of Death”

**DOI:** 10.3390/pharmaceutics15041038

**Published:** 2023-03-23

**Authors:** Ana Brotons-Canto, Claudia P. Urueña, Izaskun Imbuluzqueta, Edurne Luque-Michel, Ana Luisa Martinez-López, Ricardo Ballesteros-Ramírez, Laura Rojas, Susana Fiorentino

**Affiliations:** 1Nucaps Nanotechnology S.L., 31110 Noain, Spain; 2Grupo de Inmunobiologiay Biología Celular, Facultad de Ciencias, Unidad de Investigación en Ciencias Biomédicas, Pontificia Universidad Javeriana, Bogotá 110231, Colombia; 3DreemBio S.A.S., Bogotá 111015, Colombia

**Keywords:** nanoparticles, bioavailability, casein nanoparticles, *Caesalpinia spinosa*, P2Et, immunomodulation, natural product, cancer, metastasis, polyphenols

## Abstract

P2Et is the standardized extract of *Caesalpinia spinosa (C. spinosa)*, which has shown the ability to reduce primary tumors and metastasis in animal models of cancer, by mechanisms involving the increase in intracellular Ca++, reticulum stress, induction of autophagy, and subsequent activation of the immune system. Although P2Et has been shown to be safe in healthy individuals, the biological activity and bioavailability can be increased by improving the dosage form. This study investigates the potential of a casein nanoparticle for oral administration of P2Et and its impact on treatment efficacy in a mouse model of breast cancer with orthotopically transplanted 4T1 cells. Animals were treated with either free or encapsulated oral P2Et orally or i.p. Tumor growth and macrometastases were evaluated. All P2Et treatments significantly delayed tumor growth. The frequency of macrometastasis was reduced by 1.1 times with P2Et i.p., while oral P2Et reduced it by 3.2 times and nanoencapsulation reduced it by 3.57 times. This suggests that nanoencapsulation led to higher doses of effective P2Et being delivered, slightly improving bioavailability and biological activity. Therefore, the results of this study provide evidence to consider P2Et as a potential adjuvant in the treatment of cancer, while the nanoencapsulation of P2Et provides a novel perspective on the delivery of these functional ingredients.

## 1. Introduction

Cancer is a significant public health problem and one of the major causes of morbidity and mortality worldwide [1]. The World Health Organization (WHO) pointed out that at least 18 million new cases were diagnosed in 2018, and around 9 million deaths were caused by cancer during that year [2].

In this context, natural products have acquired great importance in tumor treatment due to their antioxidant, antiproliferative, antiangiogenic, and immunomodulatory abilities [1,3,4,5]. These abilities may be related to the prevention of carcinogenesis, the destruction of tumor cells, the modulation of the tumor microenvironment, the activation of the specific anti-tumor immune responses, or the induction of epigenetic changes, but also the improvement of the quality of life of the patients [6,7,8].

Nevertheless, the in vitro proven health benefits of these compounds do not show a clear in vivo correlation [9,10,11]. This phenomenon has been attributed to the physicochemical characteristics of natural molecules such as flavonoids or polyphenols. Generally, natural active ingredients show low water solubility, low instability in the gastrointestinal tract, and rapid metabolism. As a result, this kind of substance shows low oral bioavailability [10,12,13,14]. In addition, it must be noticed that the antitumoral activity of phytochemicals is estimated to occur through the combination of various phytochemicals acting synergistically rather than by isolated molecules [6].

To overcome this lack of in vivo efficacy, different strategies are being studied. In this regard, some efforts have explored the use of concomitant administration of the target phytochemical and metabolism inhibitors, such as piperine [15,16], or the use of synergistic combinations such as two different phytochemicals (e.g., quercetin-curcumin [10], resveratrol-curcumin [17] or curcumin-genistein [18]. Alternative strategies are the modification of the chemical structure of the phytochemical [19,20] or the design of innovative drug delivery systems [21]. However, the adequacy of human use of the ingredients in this kind of formulations and their techno-economic feasibility still needs to be clarified [22,23].

*C. spinosa* also known as Divi-divi has been traditionally used by Latin American indigenous people as a traditional medicine due to its antimicrobial, antioxidant, antitumor, and immunomodulatory properties [24,25,26,27]. The standardized extract from *C. spinosa*, called P2Et, has shown great ability to decrease lipid peroxidation and tissue damage and induce complete autophagy in tumor or stressed cells [28,29,30]. The antioxidant ability of P2Et is related to the high proportion of galloylquinic acid derivatives, and pentagalloyl glucose, among other gallic acid-containing compounds (gallates) in lower proportions [31]. Thus, P2Et acts as a potent antioxidant with high cytotoxic activity against tumor cells, especially those expressing drug resistance pumps as PgP [30]. This potential has been proved to be effective in different kinds of cancer murine models, such as breast cancer and melanoma, in which P2Et promotes the activation of the immune system, particularly T cells and production of IFN-gamma, TNF-alpha, IL-4, and IL-5, which take part on primary tumor control [32,33].

The oral administration of P2Et is safe in healthy humans with a maximum tolerated dose of 600 mg/day. At this high dose no severe toxicity has been observed. However, some gastrointestinal symptoms have been described. Even though, they do not show significant changes in safety parameters [24]. The use of gastro-resistant capsules has made it possible to solve part of these effects and demonstrate the beneficial effect of this phytomedicine in patients with COVID-19 [28]. However, it is possible to improve the pharmaceutical form of this preparation to favor its better use in patients.

In this context, this work aimed to prepare, characterize, and evaluate a P2Et-loaded nanoparticulate formulation based on casein protein and to study in vivo its oral efficacy in a murine model of breast cancer.

## 2. Materials and Methods

### 2.1. Chemicals

Sodium caseinate was purchased from Acros Organic (Geel, Belgium). L-lysine, mannitol, sodium hydroxide, hydrochloric acid, potassium phosphate monobasic, sodium chloride and Tween 20, were obtained from Sigma-Aldrich (Steinheim, Germany). Calcium chloride was from Merk (Darmstadt, Germany). Acetonitrile, methanol, and formic acid were purchased from Panreac (Barcelona, Spain). Gallic acid and ethyl gallate standards were from Sigma-Aldrich (Germany). Ethanol was obtained from OPPAC (Noain, Spain). A water purification system prepared deionized reagent water (18.2 MO resistivity) (Barbatáin, Wasserlab, Spain). Brilliant green bile broth (BGBB), violet red bile lactose agar, MacConkey Agar, Rappaport Vassiliads broth, Agar XLD, Agar cetrimide and Chapman Broth were purchased from Condalab (Torrejón de Ardoz, Spain). All reagents and chemicals used were of analytical grade. The brilliant green bile broth (BGBB), violet red bile lactose agar, MacConkey Agar, Rappaport Vassiliads broth, Agar XLD, Agar cetrimide and Chapman Broth were purchased from Condalab (Torrejón de Ardoz, Spain). Cell culture reagents (RPMI-1640, heat-inactivated fetal bovine serum (FBS), glutamine, penicillin, streptomycin, HEPES buffer, sodium pyruvate, trypsin/1X EDTA) were obtained from Eurobio Scientific, Paris, France.

Pods of *C. spinosa* (Feuillée ex Molina) Kuntze (Divi-divi or tara) were collected in Villa de Leyva, Boyacá, Colombia and identified by Carlos Alberto Parra of the National Herbarium of Colombia (copy number of voucher COL 588448). The P2Et was produced as described from the pods of *C. spinose* under GMP conditions [30]. The quality control of P2Et was performed according to the regulatory guideline 1156 of 2018 of herbal drugs in Colombia, WHO guideline “Quality control methods for herbal materials” and FDA guideline “Botanical Drug Development Guidance for Industry” [34].

### 2.2. Optimization of P2Et Encapsulation

Casein nanoparticles were prepared using a simple coacervation procedure followed by a purification step with ultrafiltration and subsequent drying by Spray-drying (see Figure 1).

Briefly, 250 mg casein and 23 mg L-lysine were dissolved in 19 mL of water type II under magnetic stirring at room temperature. Simultaneously, P2Et was dissolved in an 96% ethanol, in order to obtain a 10 mg/mL solution. Then, different volumes of the P2Et solution were added to the casein-lysine water solution until a final concentration of 3%, 4.79% and 9.14% of P2Et/total solids was obtained. Then, the solution containing casein, L-lysine and the different P2Et concentrations was incubated under magnetic stirring at room temperature.

Casein nanoparticles were obtained by adding 12 mL of a calcium chloride solution (0.2% *w*/*v*) in purified water. Afterwards, and when needed, 2 mL of mannitol solution (100 mg/mL) was added to the already formed nanoparticles prior to the drying step in a Büchi Mini Spray Drier B-191 apparatus (Büchi Labortechnik AG, Flawil, Switzerland) under the following experimental conditions: (i) Inlet temperature of 100 °C, (ii) Outlet temperature of 60–70 °C, (iii) air pressure of 4–5 bar, (iv) pumping rate of 7.5 mL/min, (v) aspirator of 80% and (vi) air flow at 900 mL/h.

For the identification of the different formulations, the following abbreviations were used: NPC-P2Et-3% (casein nanoparticles containing a theoretical concentration of P2Et of 3% (*w*/*w*)), NPC-P2Et-5% (casein nanoparticles containing 5% (*w*/*w*) theoretical concentration of P2Et), and NPC-P2Et-9% (casein nanoparticles containing 9% (*w*/*w*) theoretical concentration of P2Et).

Empty nanoparticles were prepared following the protocol in the absence of P2Et. Those particles were identified as NPC (empty casein nanoparticles).

### 2.3. Physico-Chemical Characterization of Nanoparticles

#### 2.3.1. Size, Zeta Potential and Yield

The particle size, polydispersity index (PDI) and zeta-potential (ζ) were determined with photon correlation spectroscopy (PCS) and electrophoretic laser Doppler anemometry, respectively, using a Zetasizer analyzer system (Brookhaven Instruments Corporation, Holtsville, New York, NY, USA). The diameter and polydispersity index (PDI) of the nanoparticles were determined after dispersion in ultrapure water (1/10) and measured at 25 °C with dynamic light scattering angle of 90°. The zeta potential was determined after diluting 2 mg of the sample in 2 mL of ultrapure water.

The yield of the preparative process of nanoparticles was calculated with gravimetry [35].

#### 2.3.2. P2Et Quantification

The concentrations of P2Et were extrapolated from the gallic acid and ethyl gallate concentration. P2Et quantification was determined with reverse-phase high-performance liquid chromatography (HPLC) with UV detection. The chromatographic separation was performed using a liquid chromatographic system equipped with an Alliance 2695 system connected to a Waters 2998 photodiode array detector (Waters Corp., Mildford, MA, USA).

Detection was carried out at 274 ± 4 nm and frequency of acquisition 2.5 Hz. Chromatographic separation was achieved using a Luna Omega C18 (150 × 2.1 mm; particle size 1.6 µm) column (Waters Corp., Mildford, MA, USA). The column temperature was 35 °C, and samples were stored at 15 °C before injection.

The mobile phase, pumped at 0.37 mL/min, was a gradient mixture (Table 1) of formic acid 0.1% and acetonitrile (ACN). Injection volume was set at 10 µL.

Under these conditions, the peaks for gallic acid and ethyl gallate appear at run-times of 3.05 min and 11.80 min, respectively.

For the quantification of P2Et in the nanoparticles, 30 mg of the formulation was dissolved in 10 mL of a mixture of methanol:DMSO (4:1; *v*/*v*). Each sample was analyzed in triplicate, and the results expressed as the amount of each compound per mg of the formulation.

Afterward, the encapsulation efficiency (E.E.) was calculated as follows:EE (%) = (P2Et quantified/P2Et theoretical) × 100

### 2.4. In Vitro Release Study

Release experiments were conducted at 37 °C using simulated fluids for gastric (SGF; pH 1.2) and intestinal (SIF; pH 6.8) conditions prepared according to European Pharmacopoeia (EMA, European Pharmacopoeia 6.0, chapter 2.9.3 Dissolution Test for Solid Dosage Forms, 2008).

At different intervals, samples were collected and centrifuged at 10,000 rpm for 10 min (Centrifuge MIKRO 220, Hettich, Germany). For each specific time interval, 79 μg of P2Et, either free or entrapped into nanoparticles, was resuspended in 2 mL of the corresponding simulated fluid in polyvininyl chloride tubes and maintain under constant shaking at 1500 rpm at 37 °C (Labnet VorTemp 56 EVC, Labnet International, Inc.). The different formulations were kept in the SGF for 2 h before being transferred to SIF for 20 h. The amount of P2Et released was quantified using HPLC from the supernatants described above.

### 2.5. Microbiology Evaluation of the Nanocapsules

The microbiology evaluation was performed according to European regulation (EU Regulation 2073/2005 and 1441/2007) [36] and following validated procedures from ISO standards. In addition, the acceptance criteria from USP <2023> were considered.

The total aerobic microbial count was performed according to ISO 4833:2013. The total coliforms were studied according to the ISO 4832:2006 procedure, whereas fecal coliform was performed according to ISO 4832:2006. Total yeast and molds was performed following ISO 21527:2008. The *Escherichia coli* study was conducted according to ISO 16654:2001. The *Salmonella* sp. was performed according to ISO 6579:2017, and *Staphylococcus aureus* was conducted according to ISO 6888 procedure.

### 2.6. In Vivo Evaluation of P2Et Encapsulation

#### 2.6.1. Tumor Cell Line and Culture Conditions

4T1 cells were cultured in RPMI-1640 supplemented with 10% heat-inactivated fetal bovine serum (FBS), 2 mL-glutamine, 100 U/mL penicillin, 100 mg/mL streptomycin, 0.01 M HEPES buffer, and 1 mM sodium pyruvate and incubated in a humidified environment at 37 °C and 5% CO_2_. Cells were grown until 75% confluency and passaged using trypsin/1X EDTA, washed with PBS, and resuspended in supplemented RPMI-1640 [37].

#### 2.6.2. Animals and Housing Conditions

Young female BALB/c mice (6 to 12 weeks old) were purchased from the Jackson Laboratories (Bar Harbor, ME, USA) and housed at the animal facilities of the Pontificia Universidad Javeriana (PUJ, Bogotá, Colombia) following the established protocols of the Ethics Committee of the Faculty of Sciences, PUJ, and National and International Legislation for Live Animal Experimentation (Colombia Republic, Resolution 08430, 1993; National Academy of Sciences, 2010). The present study was approved by the ethics committee of the Faculty of Sciences, PUJ, on August 9, 2018 (FUA-093-20). Mice were housed in polyethylene cages with food and water provided *ad libitum*, on a 12-h light/dark cycle at 20–22 °C and 40–60% humidity.

#### 2.6.3. In Vivo Tumor Model and P2Et Treatment

For breast cancer murine model, 1 × 10^4^ viable 4T1 cells were s.c. injected into the right mammary fat pad of BALB/c mice. To evaluate the effect of treatments on tumor growth, 3–5 days after tumor cell inoculation, eight mice per group were treated with 18.7 mg/kg P2Et (Intraperitoneally, i.p.), 18.7 mg/kg P2Et (Oral), 168.3 mg/Kg NPC-P2Et 9%, 56.1 mg/Kg NPC-P2Et 9% (Oral) or PBS (negative control) two times per week. In all experimental settings, the size of the tumors was assessed three times per week with Vernier calipers, and the volume was calculated according to the formula V (mm3) = L (major axis)2 × (minor axis)/2 [38]. Mice were euthanized by CO_2_ inhalation, and then spleen, tumor draining lymph nodes (TDLN), and tumor were removed and processed. Looking for sufficient statistical power adjusted to the standard deviation and to the proportion of losses in each model, eight mice were included for each treatment group.

### 2.7. Stability Study

For this purpose, nanoparticles and control samples were stored in 5 mL capacity glass containers in a climatic Test Cabinet TK 600 NUVE (2012) at 40 °C and 75% Relative Humidity (RH). The formulation’s mean size, PDI and P2Et content was evaluated over time.

### 2.8. Statistical Analysis

The physico-chemical characteristics of nanoparticles and the in vitro studies were compared using the Student’s *t* test. Calculations were performed using OpenEpi, version 3, (open-source calculator t_testMean). For in vivo studies, comparisons were performed using the two-way analysis of the variance (ANOVA) and Dunnett´s multiple comparison test. Data from the in vivo analysis was analyzed using GraphPad Prism v9.3.1 for Mac OS X (GraphPad Software, La Jolla, CA, USA). Data are expressed as the mean ± standard deviation (S.D.) of at least three experiments. In all cases *p* < 0.05 was considered as a statistically significant difference.

## 3. Results

### 3.1. Preparation of P2Et Nanoparticles

Although casein nanoparticles can be easily prepared using simple coacervation in the presence of calcium, the resulting nanoparticles display low stability in an aqueous environment. Thus, the basic amino acid lysine was added in order to increase the stability of casein nanoparticles following [39].

Table 2 summarizes the main physico-chemical properties of P2Et-loaded nanoparticles. When the phytotherapeutic P2Et was encapsulated into casein nanoparticles, a moderate decrease in the mean size of the resulting carriers was observed for those formulations containing lower amounts of P2Et (about 226 nm for empty nanoparticles vs. 140 nm for NPC-P2Et-3% and NPC-P2ET-5%), whereas high payload formulation shows a slightly increase in the particle size (226 nm vs. 260 nm). Similarly, the zeta potential of the particles decreased with an increase of the payload. This result may reflect the presence of P2Et on the surface of the particles.

Finally, the P2Et concentration of the most promising prototypes (NPC-P2Et-5% and NPC-P2Et-9%) was quantified (Table 3):

### 3.2. In Vitro Release Studies

P2Et release kinetics from casein nanoparticles was evaluated in two different media: Simulated gastric fluid (SGF) and simulated intestinal fluid (SIF). Figure 2 represents the release profiles of gallic acid and ethyl gallate from the formulations as a cumulative percentage of drug released as a function of time. For control purposes, free P2Et was also incubated in both fluids and the content of gallic acid and ethyl gallate was studied.

When gallic acid was directly incubated (P2Et free form) for two hours in SGF (pH 1.2) the molecule was stable. However, when the gallic acid was incubated into SIF (pH 6.8) a high degradation phenomenon was observed. In fact, only about 2–3% of the initial gallic acid was quantified at this point.

When the particles were incubated for two hours in SGF, about 70% of the total gallic acid was released. Then, when added to SIF the nanoparticles still released gallic acid. It can be observed that the total estimation of gallic acid release from the nanoparticle formulation was higher than the initial amount. This increase in the release of P2Et from the nanoparticles-based system was related to the ability of the particle to protect ethyl gallate and potentially, methyl gallate from degradation in the SGF, which may be related to an increase of the initial amount of the gallic acid.

The ethyl gallate concentration decreased by about 76% after two hours of incubation in SGF. Similarly, methyl gallate concentrations showed a reduction of about 32% in these two hours (data not shown).

Conversely, the ethyl gallate released from the nanoparticles maintained a continuous pattern, in which about 27% of the total content was released from particles as ethyl gallate.

### 3.3. Microbiological Evaluation of the Nanoparticles

The microbiology evaluation was performed according to European regulation (EU Regulation 2073/2005 and 1441/2007) [36], following validated procedures from ISO standards and considering the acceptance criteria from USP <2023>. Table 4 summarizes specifications according to the different ISO and the obtained result.

### 3.4. In Vivo NPC-P2Et Treatment Delays Tumor Growth and Metastasis in Breast Cancer Model

We have previously described that P2Et I.P treatment post-tumor engraftment delays melanoma and breast cancer tumor growth [32,33,40]. We wanted to assess whether NPC-P2Et had the same antitumor effect in the metastatic breast cancer model or if an increased activity might be observed, and if oral administration might induce the same activity previously observed for P2Et-I.P inoculation. To answer this question, we treated mice with P2Et i.p., P2Et Oral, NPC-P2Et Oral or PBS (negative control) post-tumor engraftment and evaluated the tumor growth and macro-metastasis. The mice treatment with P2Et i.p., Oral P2Et or Oral NPC-P2Et significantly delayed tumor growth compared to mice treated with PBS (Figure 3A–F). Interestingly, while the frequency of macro-metastasis was only 1.1 times reduced in mice treated with P2Et I.P treatment (Figure 3H), the highest Oral NPC-P2Et preparation (Figure 3J,K) reduced it by 3.57 times compared with PBS (Figure 3G), while Oral P2Et (Figure 3I), reduces it by 3.2 times. This result suggests that nanoencapsulation led to the administration of higher doses of effective P2Et by the oral route, slightly improving bioavailability and biological activity, principally evidenced by the control of metastasis, and possibly decreasing gastrointestinal side effects due to the characteristics of the nanoparticles used to protect P2Et. In conclusion, the mice treated with higher doses of NPC-P2Et showed a low incidence of metastatic cells (Figure 3J,K).

### 3.5. Stability Study

The stability of nanoparticles was estimated over time considering modification on the mean size, PDI and P2Et content of the formulation. The size and PDI of the formulation for 4.4 months show that the mean size of the particles was stable, and the polydispersity index was always below 0.3. This value was considered as an acceptable limit for PDI (see Table 5).

The P2Et concentration over time was estimated according to the gallic acid and ethyl gallate concentration in the formulation over time. During the stability test the concentration of ethyl gallate remained stable, whereas the concentration of gallic acid slightly varied as shown in Figure 4.

## 4. Discussion

The oral administration of natural compounds such as P2Et need to face several biological barriers (pH, low water solubility, high degradation through the harsh conditions of the gastrointestinal tract) that make the oral bioavailability of the drug almost negligible. From a theoretical point of view, the use of nanoencapsulation of such compounds may be an adequate approach to reach a successful oral administration as the encapsulation will protect the loaded cargo against premature degradation, including the acidic conditions of the stomach, the presence of enzymes, as well as the mechanical stress in the lumen (e.g., osmotic pressure and peristalsis) [41,42].

From a general point of view, protein-based nanoparticles offer some advantages for drug delivery purposes, including their biodegradability and capability to accommodate a high variety of compounds in a non-specific way [43]. In this context, casein possesses some characteristics that may be considered as additional advantages for the formulation of nanoparticles as its regulatory status and low toxicity [44].

In this work, casein-based nanoparticles were selected to evaluate and compare their ability to deliver P2Et by the oral route. Both types of nanoparticles (empty and P2Et-loaded nanoparticles) displayed a mean size of around 250 nm (226 nm for empty nanoparticles and 260 nm for P2Et-loaded nanoparticles) and negative zeta potential (Table 2). Notably, the surface zeta potential of P2Et-loaded nanoparticles was remarkably less negative than that of empty nanoparticles, suggesting that, in the formation of P2Et-loaded nanoparticles, some structural changes may occur during P2Et encapsulation, and some P2Et molecules can adhere to the nanoparticle surface, inducing the observed changes with respect to the zeta potential. The most promising P2Et formulations (NPC-P2Et-9%) had the ability to be entrapped, with high encapsulation efficiency, by both the main components of P2Et (gallic acid and ethyl gallate). The release of P2Et from the nanoparticles appears to modify its own degradation pattern under gastrointestinal conditions. Thus, non-encapsulated gallic acid is highly degraded in simulated intestinal fluid, while encapsulation seems to protect it against this degradation, allowing its detection even after 2–6 h of incubation in this fluid. Similarly, ethyl gallate concentrations appear to be released in a continuous pattern over the time of the experiment. In fact, the benefit of casein nanoencapsulation in the preparation of new forms of drug delivery has been previously reported [45], and, particularly with polyphenols, nanoencapsulation prolongs stability and improves release [46]. In this sense, the efficacy of the formulation has been evidenced with an in vivo murine breast cancer model. Mice treated with both PC-P2Et preparations significantly reduce the size of the primary tumor, but the high concentration is more effective in reducing metastatic foci (Figure 3).

In addition, our formulation offers some interesting advantages that may enable a translational approach; particularly in those aspects related to the scale-up of a reproducible process (including the drying step) and the simplification of non-clinical toxicity assessments of the regulatory dossier. Among others, P2Et-loaded casein nanoparticles may be obtained in a simple and scalable preparative process that only requires the use of simple reagents and solvents (ethanol and water) and allows the generation of a powder formulation, easily dispersible in water. Furthermore, casein nanoparticles do not enter the systemic circulation, minimizing a possible accumulation in the body and toxicological issues [44], and for the time studied the physico-chemical characteristics of the formulation (size and PDI) are not modify over time, whereas active ingredient concentration variation was lower than 10%.

## 5. Conclusions

In this work, casein nanoparticles containing P2Et were satisfactorily prepared. These particles (NPC-P2Et) exhibited high EE for the active compounds of P2Et (gallic acid and ethyl gallate). According to the in vitro release study, this nanoparticles-based system modifies the pattern of release/absorption of the compounds in the gastrointestinal tract and thus, their bioavailability. Therefore, nanoencapsulated P2Et can improve the absorption profile of P2Et compounds, resulting in a greater therapeutic response. Our results agree that polyphenols, particularly P2Et, can be used within the framework of antitumor therapy, with an increase in the treatment efficiency when P2Et is nanoencapsulated.

These particles have a preparation procedure that only requires natural, non-toxic ingredients and are not subjected to nanomaterial restrictions. Finally, the results of this study provide motivating elements to consider P2Et as a potential adjuvant in cancer treatment. However, additional studies are required to confirm the results of this study and whether the encapsulation of P2Et in casein-based formulation provides a novel insight into the delivery of these functional ingredients due to its preclinical history.

## Figures and Tables

**Figure 1 pharmaceutics-15-01038-f001:**
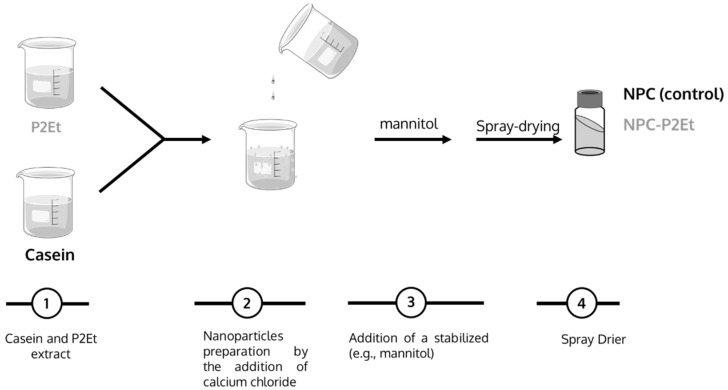
Schematic representation of the preparative process of NPC (control; empty casein nanoparticles) and NPC-P2Et (P2Et-loaded casein nanoparticles).

**Figure 2 pharmaceutics-15-01038-f002:**
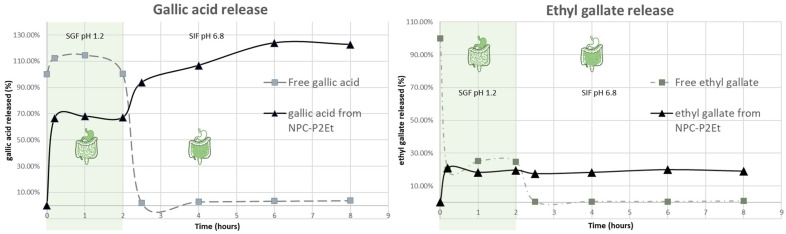
Behavior of P2Et in simulated gastric fluid and simulated intestinal fluid. The left graph shows the concentration over time of gallic acid in both fluids in the free P2Et form (square and grey line) and the encapsulated form (triangle and black line). On the right graph the behavior of ethyl gallate over time is shown. The free P2Et form is shown as a square and grey line while the encapsulated form is shown as a triangle and black line. Results from *n* = 1 test.

**Figure 3 pharmaceutics-15-01038-f003:**
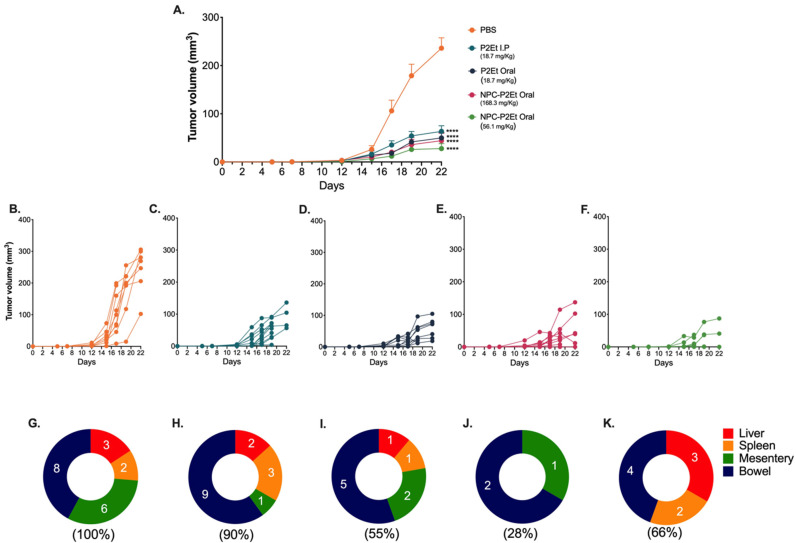
NPC-P2Et delay tumor growth and macro-metastasis in breast cancer model. 4T1 tumor growth curve after treatments (**A**). 4T1 individual tumor volume after PBS (**B**), P2Et I.P (18.7 mg/Kg) (**C**), P2Et Oral (18.7 mg/Kg) (**D**), NPC-P2Et Oral (168.3 mg/Kg), (**E**), NPC-P2Et Oral (56.1 mg/Kg) treatment (**F**). Distribution of multi-organ metastasis of 4T1 tumors for all groups, after PBS (**G**), P2Et I.P (18.7 mg/Kg) (**H**), P2Et Oral (18.7 mg/Kg) (**I**), NPC-P2Et Oral (168.3 mg/Kg), (**J**). NPC-P2Et Oral (56.1 mg/Kg treatment (**K**). In all cases, data are represented as the mean ± SEM. The *p* values were calculated using a Dunnett´s multiple comparison test. **** *p* < 0.0001.

**Figure 4 pharmaceutics-15-01038-f004:**
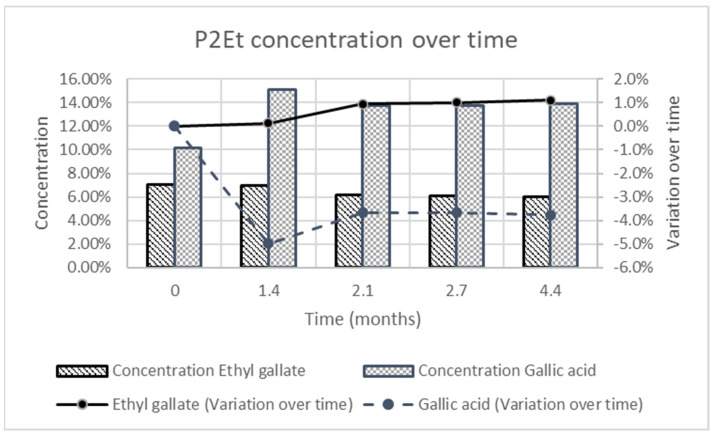
Variation of P2Et concentration over time according to gallic acid and ethyl gallate concentration over time. The standard deviation of ethyl gallate variation over time were 0.19%, 0.38%, 1.53%, 1.61% and 1.76% at the different times (0, 1.4, 2.1, 2.7 and 4.4 months), whereas for the same time the standard deviation of gallic acid variation over time were 0.01%, 3.63%, 3.26%, 3.30% and 3.46%.

**Table 1 pharmaceutics-15-01038-t001:** Composition and gradient of the mobile phase required for P2Et estimation using HPLC.

Time (Minutes)	Formic Acid 0.1% in Water (%)	ACN (%)
0	99	1
6	99	1
11	40	60
16	40	60
26	99	1

**Table 2 pharmaceutics-15-01038-t002:** Physico-chemical properties of the casein-based nanoparticles.

Sample	Mean Size (nm)	PDI	Zeta Potential (mV)	Theoretical [P2Et] (%)
NPC (control)	226 ± 6	0.189 ± 0.013	−17.6 ± 0.3	-
NPC-P2Et-3%	145 ± 2	0.122 ± 0.014	−13.6 ± 6.4	3.00
NPC-P2Et-5%	146 ± 4	0.133 ± 0.031	−9.5 ± 3.4	4.79
NPC-P2Et-9%	260 ± 13	0.116 ± 0.011	−9.4 ± 0.9	9.14

**Table 3 pharmaceutics-15-01038-t003:** P2Et concentration into nanoparticles. The concentration of P2Et entrapped into nanoparticles is based on the quantification of two different molecules: gallic acid and ethyl gallate.

Sample	Gallic Acid Payload (mg g.a./mg NP)	Gallic Acid Encapsulation Efficiency (%)	Ethyl Gallate Payload (mg e.g.,/mg NP)	Ethyl Gallate Efficiency (%)
NPC-P2Et-5%	31.77	66	35.08	100
NPC-P2Et-9%	92.56	100	64.73	100

**Table 4 pharmaceutics-15-01038-t004:** Microbiological results of the obtained formulation estimated according to ISO recommended procedures.

Microorganism	Specifications	Result	Methods
Aerobic mesophilic bacteria (CFU/g)	Max 100.000	4.000	ISO 4833:2013
Total coliform (CFU/g)	Max 10	<10	ISO 4832:2006
Fecal coliform (CFU/g)	Max 10	<10	ISO 7251:2005
*Escherichia coli* (CFU/10 g)	Max 10	<10	ISO 16654:2001
*Staphylococcus aureus* (CFU/g)	Max 10	<10	ISO 6888
*Pseudomona aeruginosa* (CFU/10 g)	Max 10	<10	ISO 22717:2015;
*Salmonella* spp.	Max 10	<10	ISO 6579:2017

**Table 5 pharmaceutics-15-01038-t005:** Mean size and PDI of the formulation over time.

Time (Months)	Mean Size (nm)	PDI
0	202 ± 13	0.208 ± 0.073
1.4	193 ± 3	0.248 ± 0.024
2.1	215 ± 28	0.272 ± 0.004
2.7	224 ± 16	0.287 ± 0.019
4.4	208 ± 16	0.250 ± 0.080

## Data Availability

Data will be made available on a reasonable request.

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
