# Peer review of "Encapsulated Phytomedicines against Cancer: Overcoming the “Valley of Death”"

_pharmaceutics, 2023, doi:10.3390/pharmaceutics15041038_

Round 1

Reviewer 1 Report

Reviewer

Encapsulated Phytomedicines Against Cancer: Overcoming the "Valley of Death".

The Manuscript is interesting, very well organized. But the introduction in very general without an explaining more strongly the problem to be solved.

Some phrases are translated from Spanish to English. All my comments are the manuscripts with comments.

For example: There are spaces between words and all words in vitro must be in italics.

Author Response

We would like to thank the reviewer the feedback, All suggested changes found in the manuscript have been included. Also the formatting issues have been addressed.

Reviewer 2 Report

The manuscript “Encapsulated Phytomedicines Against Cancer: Overcoming the "Valley of Death"” discussed the potential of casein nanoparticles for oral administration of P2Et and its effect on the therapy of orthotopic breast cancer mouse model. This work confirmed that nanocapsules with higher P2Et loading can display higher bioavailability and bioactivity of the drug, so that it can be better absorbed and utilized, thus providing a novel approach on the delivery of various functional ingredients. The material and method part of the whole paper is described in detail, and the experimental part is also relatively complete, but there are a few issues to be addressed before the manuscript could be published:

1.        There are some mistakes in the use of many symbols, such as , which was used a lot in the Materials method section.

2.        There are also some formatting issues, such as the use of some units. 2 milliliter should be written 2 mL, not 2 ml. It's best to pay attention to formatting uniformity.

3.        There are some careless problems, such as subheadings with an extra dot after the ordinary number (e.g. 2.3..).

4.        All the analytical data should be provided with an error bar representing the standard deviations (e.g. Figure 2 and Figure 4).

Author Response

We appreciate the suggestions of the reviewer, and the manuscript has been modifies. Celsius abbreviation has been corrected. Also all “milliliter” have been corrected and homogenized to “mL” abbreviation. The  heading and subheading have been review in order to remove extra “dots” found in some subheadings.

Finally, the information from figure 2 and 4 have been modified in order to include information regarding the standard deviations as suggested by the reviewer.

Reviewer 3 Report

Congratulations to authors !

The study is well structured and clearly presented. In fact the authors wanted to assess whether NPCP2Et had the same antitumor effect in the metastatic breast cancer model or if an increased activity might be observed, and if oral administration might induce the same activity previously observed for P2Et-I.P inoculation. They treated mice with P2Et I.P, P2Et Oral, NPC-P2Et Oral or PBS (negative control) post- tumor engraftment and evaluated the tumor growth and macro-metastasis. It was found that nanoencapsulation can slightly improve the bioavailability of P2Et, conclusion that is well supported by the presented results.

Same minor corrections need to be done (please consider the highlits marked in the text of the manuscript). Please change in the entire text in vitro and in vivo terms in italics. 

Please also clarify the conditions of dissolution tests: i.e. temperature, mixing speed.

Could we please better explain the statement: ... increase is related to the ability of the particle to protect ethyl gallate and potentially, methyl gallate from degradation in the SGF.  How this protection can induce an increase of the initial amount of the bioactive ?

Author Response

Thanks for the feedback.

The formatting issues such as the highlight and “in vitro” and “in vivo” have been modified as suggested by the reviewer.

Additional information about the dissolution test has been included. We would like to thank the reviewer the suggestion.

Finally, we have modified the statement about the protection of P2Et in SGF. Unfortunately, it was not possible to estimate the concentration of methyl gallate on the NP, which may be consider as a potential source of gallic acid, increasing the amount of final gallic acid at the end of the experiment.